# Temporal Trend, Causes, and Timing of Neonatal Mortality of Moderate and Late Preterm Infants in São Paulo State, Brazil: A Population-Based Study

**DOI:** 10.3390/children10030536

**Published:** 2023-03-10

**Authors:** Maria Fernanda B. de Almeida, Adriana Sanudo, Kelsy N. Areco, Rita de Cássia X. Balda, Daniela T. Costa-Nobre, Mandira D. Kawakami, Tulio Konstantyner, Ana Sílvia S. Marinonio, Milton H. Miyoshi, Paulo Bandiera-Paiva, Rosa M. V. Freitas, Liliam C. C. Morais, Mônica L. P. Teixeira, Bernadette Waldvogel, Carlos Roberto V. Kiffer, Ruth Guinsburg

**Affiliations:** 1Escola Paulista de Medicina, Universidade Federal de São Paulo, São Paulo 04023-062, SP, Brazil; 2Fundação Sistema Estadual de Análise de Dados, São Paulo 05508-000, SP, Brazil

**Keywords:** infant newborn, infant, premature, neonatal mortality, cause of death, developing countries, epidemiological studies

## Abstract

Moderate and late preterm newborns comprise around 85% of live births < 37 weeks gestation. Data on their neonatal mortality in middle-income countries is limited. This study aims to analyze the temporal trend, causes and timing of neonatal mortality of infants with 32^0/7^–36^6/7^ weeks gestation without congenital anomalies from 2004–2015 in the population of São Paulo State, Brazil. A database was built by deterministic linkage of birth and death certificates. Causes of death were classified by ICD-10 codes. Among 7,317,611 live births in the period, there were 545,606 infants with 32^0/7^–36^6/7^ weeks gestation without congenital anomalies, and 5782 of them died between 0 and 27 days. The neonatal mortality rate decreased from 16.4 in 2004 to 7.6 per thousand live births in 2015 (7.47% annual decrease by Prais–Winsten model). Perinatal asphyxia, respiratory disorders and infections were responsible, respectively, for 14%, 27% and 44% of the 5782 deaths. Median time to death was 24, 53 and 168 h, respectively, for perinatal asphyxia, respiratory disorders, and infections. Bottlenecks in perinatal health care are probably associated with the results that indicate the need for policies to reduce preventable neonatal deaths of moderate and late preterm infants in the most developed state of Brazil.

## 1. Introduction

Prematurity was, globally, the leading cause of death among children younger than 5 years in 2019. Among 5.3 million deaths (95% uncertainty range 4.92–5.68) of children younger than 5 years, preterm birth complications were responsible for 17.7% (95%UR 16.1–19.5) [1]. In high-income countries, the challenge is survival of extremely preterm infants (<28 weeks gestation) without neurodevelopmental impairments [2,3], with an increasing interest in nano-preterm infants of 22–23 weeks gestation [4]. In low- and middle-income countries, around 80% of newborn deaths occur among low-birth-weight infants, especially those born preterm [5,6]. Deaths of infants with a gestational age of 32–36 weeks [7,8] have a special interest because they are potentially preventable [9].

Rates of preterm birth vary worldwide from 7 to 14%, but in all countries, the major contributors are births of moderate and late preterm infants [8]. In 2020, in the USA, live births with 32–36 weeks gestation comprised 85% of the 364,487 live births under 37 weeks gestation [10]. Moderate and late preterm infants are both physiologically and developmentally immature and have a higher risk for morbidity compared with infants born at term [7], including difficulties in the cardiopulmonary transition at birth, respiratory disorders, hypoglycemia, jaundice, infections, and delayed breastfeeding, along with adverse neurodevelopmental outcomes and death [11,12,13].

In 2020, in Brazil, considered an upper-middle-income country [14], among 2,730,145 live births, 308,702 were preterm, and 86% of these had a gestational age of 32–36 weeks. Moderate and late preterm infants were responsible for 17% of neonatal deaths in 2020 in the country, with a neonatal mortality rate of 12.5 per thousand live births [15]. Among the 27 federative units of the country, São Paulo (Southeast region of Brazil) is the most populous and richest state. It has around 47 million inhabitants, comprising 22% of the Brazilian population [16]. The state has a Human Development Index of 0.806 (2021) [17], but there are important social and economic inequalities, as frequently seen in middle-income countries [18]. The study of neonatal mortality of infants with 32–36 weeks gestation in São Paulo State, who have more than 98% chance of survival in high-income countries [19], may help to understand the bottlenecks that need to be overcome in order to increase equity in healthcare. According to Marmot, there are positive reasons for working with cities and regions on action to achieve health equity: the local context is where people are born, grow, live, work and age. There is more understanding of the reality of people’s lives and the actions necessary to change the social determinants of health [20].

This study’s aim is to evaluate the temporal trend, causes and timing of neonatal mortality of infants with 32^0/7^–36^6/7^ weeks gestation without congenital anomalies in São Paulo State, Brazil, from 2004 to 2015.

## 2. Materials and Methods

This population-based study enrolled all live births with 32^0/7^–36^6/7^ weeks gestation without congenital anomalies in São Paulo State, Brazil, from 2004 to 2015. The Statistical Bureau of the State of São Paulo (Fundação SEADE) provided the database retrieved from the Civil Registry information, which covers 99.7% of live births and 99.8% of deaths in the state [21].

Fundação SEADE provided two different sets of data: (1) information on all live-born infants from 2004–2015 originated from live birth certificates; (2) information on all infant deaths during the same period, originated from death certificates linked to live birth certificates by deterministic linkage, as previously detailed [22]. Briefly, the database used for the study was obtained by linking the two data sets above using the birth variables present in both of them. These steps were performed twice for 2004–2013 and for 2014–2015 live births.

Infants were classified as alive if they survived the neonatal period or as neonatal deaths if occurring before 28 days after birth. The gestational age of 32^0/7^–36^6/7^ weeks, as reported in the live birth and/or death certificate, was considered for inclusion. The exclusion criteria were the presence of congenital malformations reported in Live Birth and/or Death Certificates with any Q code (International Classification of Diseases, 10th revision—ICD-10) [23], and/or unknown birth weight.

The following data were retrospectively collected: (1) birth variables: year and place; (2) maternal variables: age, civil status, education, parity, single or multiple pregnancy, prenatal care, and delivery mode; (3) neonatal variables: sex, birth weight, and 1st- and 5th-minute Apgar score. Regarding neonatal deaths, the following items were also collected: age of death and causes mentioned in any line of the death certificate. The causes of death were classified into the following groups according to death certificate codes [23]: perinatal asphyxia (P20, P21, P24.0), respiratory disorders [P22, P23, P24 (except P24.0), P25 to P28], infections (P35 to P39), and others.

Prais–Winsten model was applied to evaluate the annual trend of the neonatal mortality rate. The annual percent change (APC) with 95% confidence intervals (95% CI) was classified as stationary, increasing or decreasing in time. For non-stationary trends, Poisson regression analysis was modeled with year as the only independent variable.

The temporal trend of neonatal mortality rate associated with perinatal asphyxia, respiratory disorders, infections, and other causes was also analyzed using the Prais–Winsten model. The proportion of deaths by these causes during the study years was compared using chi-square for trend. A Kaplan–Meier curve was used to assess age at death for each of the main groups of causes.

Place of birth and maternal and neonatal variables were compared between infants who survived vs. those who died in the neonatal period. Poisson multivariate regression with backward analysis was adjusted by year of birth to evaluate variables associated with neonatal death. Data are expressed in incidence rate ratio (IRR) and 95% CI.

Software-based statistical analysis was performed using Stata/SE 17 (StataCorp, 2021. College Station, TX, USA: StataCorp LLC). The study was approved by the Ethics Committee on Human Research of Escola Paulista de Medicina—Universidade Federal de São Paulo, and by the Board of Directors of Fundação SEADE.

## 3. Results

There were 7,317,611 live births in São Paulo State, Brazil, from 2004 to 2015. Among them, 545,606 infants with 32^0/7^–36^6/7^ weeks gestation without congenital anomalies were included and 5782 died between 0 and 27 days (Figure 1), with a mortality rate 10.58 (95% CI 10.32–10.84) per thousand live births during the study period. 

The mortality rate decreased from 16.4 in 2004 to 7.6 per thousand live births in 2015 (Figure 2). The Prais–Winsten model showed an annual decrease of −7.47% (95% CI: −6.35 to −8.58%) in the neonatal mortality rate of infants with 32^0/7^–36^6/7^ weeks gestation without congenital anomalies. From 2004 to 2015, there was a decrease of 46% (95% CI 41–52%) in the neonatal mortality rate (Poisson regression analysis with robust variance; *p* < 0.001).

During the study period, the neonatal mortality rate decreased for all causes of death (Figure 3). The Prais–Winsten model showed an annual percent reduction of −7.59% (95% CI −9.95 to −5.18) for deaths associated with perinatal asphyxia; −9.66% (−10.78 to −8.53) for deaths associated with respiratory disorders; −7.07% (−9.62 to −4.44) for deaths associated with infections; and −4.18% (−6.46 to −1.84) for other causes of death. Perinatal asphyxia, respiratory disorders and infections were responsible, respectively, for 14%, 27% and 44% of the 5782 deaths. Other causes of death were responsible for 15% of the 5782 neonatal deaths: hematologic in 2.8% (mainly disseminated intravascular coagulation and hemolytic anemias), hemodynamic in 2.1% (mainly congestive heart failure), neurologic in 1.1% (mainly intracranial hemorrhage), metabolic in 0.9% (mainly renal failure), and diverse causes in 8.1%. From 2004 to 2015, the yearly proportion of deaths associated with respiratory morbidities significantly decreased (Table 1).

Information on the maternal and neonatal characteristics was available as follows: 100% for neonatal sex, birthweight, and mortality timing; 98% or more for maternal age, prenatal care, multiple/single gestation, type of delivery and hospital delivery; 78% for marital status, maternal schooling and Apgar score at 1 and 5 min; and 67% for parity. Maternal and neonatal characteristics were compared according to death or survival up to 27 days after birth (Table 2). Deliveries occurred in the hospital setting in 99% of infants that survived or died in the neonatal period. Variables independently associated with neonatal death in infants with 32^0/7^–36^6/7^ weeks gestation according to Poisson multivariate regression analysis adjusted by year of birth were maternal age <20 years, maternal schooling <12 years, prenatal care <7 visits; cesarean section; male sex; and birthweight <2500 g, and Apgar score < 7 in the 1st minute (Table 2).

Median timing of mortality for all 5782 neonatal deaths was 96 h (95%CI 96–98) after birth. The median time of death according to the cause was, for perinatal asphyxia, 24 h (24–24); for respiratory disorders, 53 h (48–66); for infections, 168 h (156–168); and others, 96 h (75–98) (Figure 4).

## 4. Discussion

The main finding of this population-based study is that the neonatal mortality rate of infants with 32^0/7^–36^6/7^ weeks gestation without congenital malformations decreased from 16.4 in 2004 to 7.6 per thousand live births in 2015, in São Paulo State, Brazil, with an annual reduction of 7.47%. Perinatal asphyxia, respiratory disorders, and infections were responsible, respectively, for 14%, 27% and 44% of the 5782 deaths. The neonatal mortality rate for each one of these causes decreased from 2004 to 2015. Maternal and neonatal demographic variables were associated with the studied deaths. A persistent 0–6 Apgar score at 1 and 5 min after birth was associated with a 20 times higher chance of neonatal death, while the recovery at 5 min to a score of 7–10 was associated with a 3 times higher chance of neonatal death, compared to infants that have 7–10 Apgar scores at 1 and 5 min.

The neonatal mortality rate of 10.58 per thousand live births of infants with 32^0/7^–36^6/7^ weeks gestation without congenital malformations in the most developed state of the country is high if we compare it with data from high-income countries. In the United Kingdom, the neonatal mortality rate per thousand live births with 32–36 weeks gestation decreased from 6.33 in 2007, to 5.24 in 2015 and to 4.67 in 2018 [24]. In the USA, the neonatal mortality was 8.57 per thousand live births with 32–36 weeks gestation in 2018 [25]. Both datasets considered live births with congenital malformations, which differs from our population-based study, which focused on preventable neonatal deaths. According to Rutstein et al. [26], preventable deaths are those that could have been prevented, in whole or in part, by effective health services. In Brazil, avoidable deaths of children under 5 years have been classified as those reducible through immunoprevention measures and adequate perinatal care for women, fetuses and newborns; adequate measures of diagnosis and treatment; and appropriate health promotion [27]. Deaths associated with perinatal asphyxia, respiratory disorders and infections are considered preventable and may be reduced by the combination of measures cited above. It is important to emphasize that the studied population had a mean birthweight of 2511 ± 554 g, which illustrates the high survival probability. All these infants could have survived if essential care was promptly offered [28].

Despite the high neonatal mortality rate of moderate and late preterm infants without congenital anomalies, there was a significant reduction during the study period, with the most impressive decline from 2004 to 2013, followed by a plateau from 2013 to 2015. The same pattern could be noted for the neonatal mortality rate associated with perinatal asphyxia, respiratory disorders and infections. These results follow the findings of Saltarelli et al. [27] in a populational study about preventable deaths of children younger than five years in the Southeast Region of Brazil from 2000 to 2013. According to these authors, the preventable causes of death had a reduction of 4.4% per year, with a 44% decrease in the studied period. Since the beginning of the 21st century, Brazil granted special attention to maternal and neonatal care, with a sequence of public policies leading to the strengthening of comprehensive perinatal care, from pregnancy to neonatal care, as a strategy progressively organized in the country [29]. The impact of these healthcare strategies probably contributed to the continuous reduction in neonatal deaths of moderate and late preterm infants without congenital anomalies reported in our study until 2013. The 2013–2015 plateau of the neonatal mortality rate of moderate and late preterm infants in general, as well as the rates of neonatal mortality associated with perinatal asphyxia, respiratory disorders, and infections, may be attributed to the economic Brazilian recession and major cuts imposed to the public health system during this period, with the disorganization of maternal and newborn programs that were previously implemented [30].

The median neonatal mortality timing was 96 h, but it varied according to the cause of death as early as 24 h for perinatal asphyxia and as late as 168 h for infections. Our results are consistent with those reported by Dol et al., in a systematic review of neonatal mortality timing of late preterm and term infants [31]. Based on data from 46 studies with 6,760,731 live births and 47,551 neonatal deaths, the highest proportion of neonatal deaths occurred on day 1 (39.5%), followed by days 2–7 (36.8%), with the remainder occurring between days 8 and 28 (23.0%). Ten studies of 1326 deaths with perinatal asphyxia and 1562 deaths with infections showed that 68% of the first ones occurred on the first day after birth and 48% of the latter occurred between days 2–7 [31]. The results of our population-based study, with 99% of in-hospital births, show that infections were responsible for 40–49% of neonatal deaths of moderate and late preterm infants without congenital malformations yearly, with a median neonatal mortality timing at the end of the first week after birth. These findings reinforce the need for investment in the quality of the healthcare system. According to Dol et al. [31], healthcare providers should be well-trained and knowledgeable and have access to life-saving equipment and medication to provide a high quality of care across the high-risk time points during the first month after birth.

Lower maternal age and schooling, lower number of prenatal care visits, cesarean section, male sex, low birth weight, and low 1st- and 5th-minute Apgar score were associated with the neonatal deaths of moderate and late preterm infants without congenital malformations. These social, maternal and neonatal variables are extensively studied as risk factors for unfavorable neonatal outcomes [5,22,32]. It is interesting to note the contribution of the low vitality at birth to these deaths: one in every four neonatal deaths of the studied infants had a persistent Apgar score of 0–6. Adequate care for mothers at delivery and skilled teams apt to provide appropriate resuscitation procedures to the neonate immediately after birth could prevent these avoidable deaths [33].

Bottlenecks in perinatal health care are probably associated with the results, which indicate the need for policies to reduce preventable neonatal deaths of moderate and late preterm infants in the most developed state of Brazil. Providing a better connection between pregnant women and reference maternities would help to improve the quality of childbirth care [34]. Another important goal is to decrease the extremely high number of C-sections in the studied region. In 2020, in São Paulo State, 64% of the 54,614 live births with 32–36 weeks gestation were delivered by C-section [15]. Infants who do not follow the physiologic birthing process and do not have a strict medical indication for a preterm delivery may suffer the consequences of a more difficult adaptation to the extrauterine environment. According to Tubbs-Cooley et al., quality improvement initiatives consistently demonstrate the positive effect of bedside nursing care on clinical and safety outcomes in neonatal intensive care units [35]. As shown in this study, infection was responsible for almost 50% of the deaths of moderate and late preterm infants, with these deaths occurring by the end of the first week after birth, suggesting the role of hospital-acquired infections. Addressing the problems related to staffing of the neonatal units in the state, with an adequate number of qualified nurses, may help to decrease these deaths.

The use of secondary data, with limited available information on clinical variables that are important in the determination of death, is a limitation of our study. The use of data derived from death certificates, based on the notification of the main causes reported by physicians, does not allow us to sort out the multiple overlapping causes associated with these deaths. The dataset did not allow us to more granularly evaluate deaths by each gestational age week, as it only contains the aggregate data on live births with 32^0/7^–36^6/7^ weeks gestation. Additionally, the database is provided by Fundação SEADE after linkage and anonymization, a manual time-consuming process. Therefore, the 2015 data set was the most recent available to study. The neonatal mortality data evaluated in this study is regional, but, as cited in the introduction, there are positive reasons for working with cities and regions on action to achieve health equity: the local context is where people are born, grow, live, work and age [20]. It should be noted that the number of deaths that were actually preventable was not studied in an individual-by-individual manner, but our results might provide basic data for future research to reduce neonatal mortality in moderate and late preterm infants.

Despite these limitations, this study is one of the first to provide a populational picture of preventable neonatal deaths of moderate and late preterm infants without congenital malformations in a region of a middle-income country. As a strength of the study, São Paulo is the only Brazilian state that has developed its own system of producing independent vital statistics, which manages to relate continuous data from the civil registry with epidemiological data originating from death and live birth certificates, allowing a consistent analysis of health indicators [36].

## 5. Conclusions

Although the neonatal mortality rate of infants with 32^0/7^–36^6/7^ weeks gestation without congenital anomalies decreased significantly from 16.4 in 2004 to 7.6 per thousand live births in 2015 in São Paulo State, Brazil, the rate is still high compared to high-income countries. The causes associated with these deaths were perinatal asphyxia, respiratory disorders and infections, which are largely preventable. Possible strategies to change the picture revealed by this study include improving prenatal care quality and a link between this care and reference maternity hospitals, decreasing elective cesarean sections and increasing the availability of qualified human resources in neonatal care.

## Figures and Tables

**Figure 1 children-10-00536-f001:**
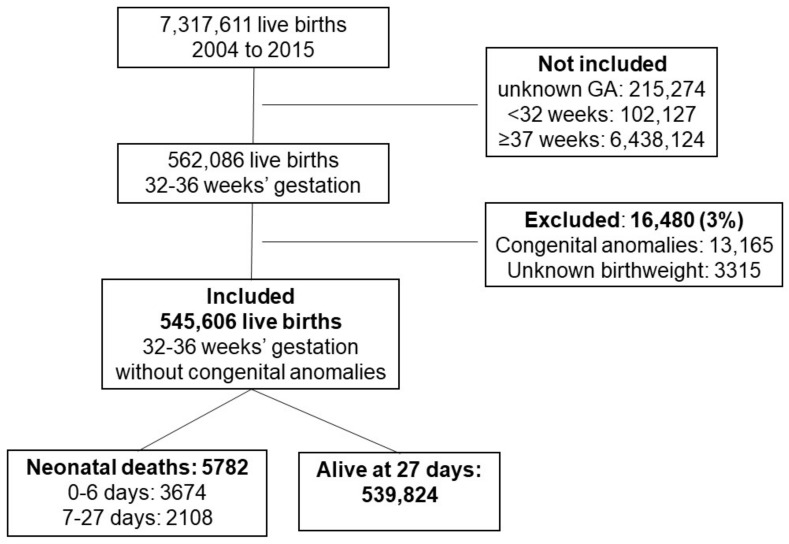
Flow diagram of the included infants in the study.

**Figure 2 children-10-00536-f002:**
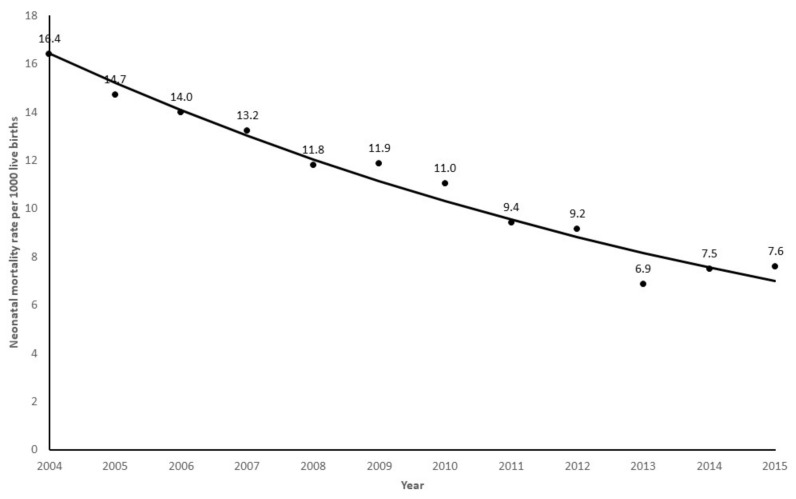
Annual trend of neonatal mortality rate of infants with 32^0/7^–36^6/7^ weeks gestation without congenital malformations adjusted with the Prais–Winsten model (• observed rate; −−− expected rate).

**Figure 3 children-10-00536-f003:**
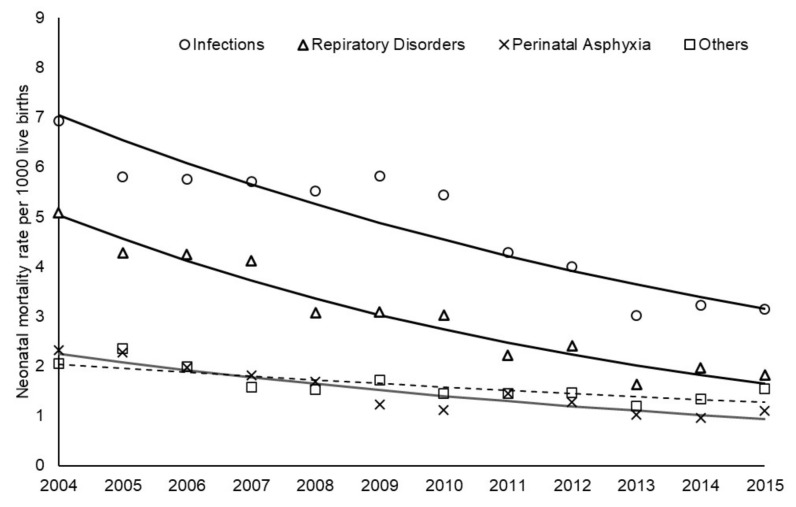
Annual trend of neonatal mortality rate of infants with 32^0/7^–36^6/7^ weeks gestation without congenital malformations adjusted by Prais–Winsten model, according to cause of death (ο, Δ, □ and × observed rate; −−− or - - - expected rate).

**Figure 4 children-10-00536-f004:**
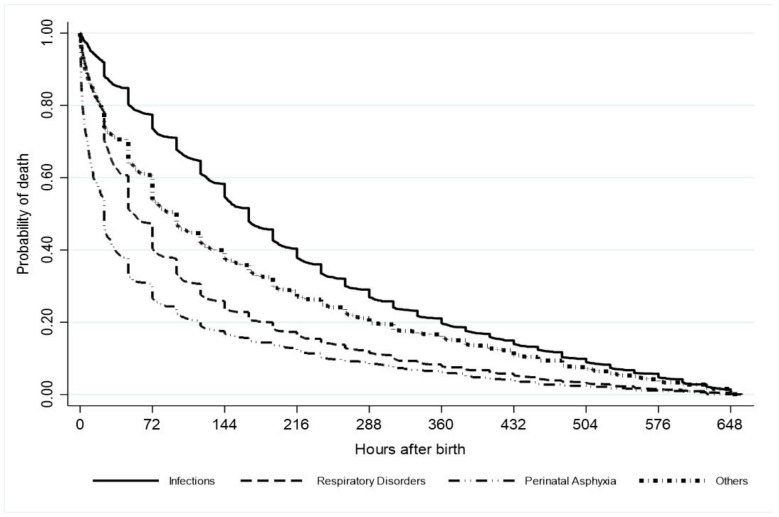
Kaplan–Meier estimate of hours after birth until death according to cause of death.

**Table 1 children-10-00536-t001:** Causes of neonatal death of preterm infants with 32^0/7^–36^6/7^ weeks gestation without congenital malformations. São Paulo State, Brazil, 2004–2015.

Year	Total Number of Deaths	Perinatal Asphyxia	Respiratory Disorders	Infections	Other
2004	606	14%	31%	42%	13%
2005	575	15%	29%	40%	16%
2006	490	14%	31%	41%	14%
2007	494	14%	31%	43%	12%
2008	441	14%	26%	47%	13%
2009	491	10%	26%	49%	15%
2010	471	10%	28%	49%	13%
2011	420	16%	24%	45%	15%
2012	516	14%	26%	44%	16%
2013	437	15%	24%	44%	17%
2014	427	13%	26%	43%	18%
2015	414	15%	24%	41%	20%
Total	5782	14%	27%	44%	15%
Chi-square for trend		*p* = 0.666	*p* < 0.001	*p* = 0.291	*p* < 0.001

**Table 2 children-10-00536-t002:** Variables associated with neonatal deaths of infants with 32^0/7^–36^6/7^ weeks gestation without congenital malformations according to Poisson multivariate regression adjusted by year. São Paulo State, Brazil, 2004–2015.

	Alive at 27 Days	Death 0–27 Days	Incidence Rate Ratio (95% CI)
Maternal age	n = 539,808	n = 5781	
<20 years	16%	21%	1.22 (1.12–1.34)
20–34 years	68%	65%	reference
≥35 years	16%	14%	0.84 (0.75–0.93)
Unmarried	n = 424,361	n = 4864	
Yes	52%	60%	1.06 (0.98–1.14)
Schooling	n = 422,885	n = 4852	
≤7 years	24%	32%	1.35 (1.20–1.52)
8–11 years	55%	56%	1.22 (1.10–1.36)
≥ 12 years	21%	12%	reference
Primiparity	n = 359,937	n = 3964	
Yes	49%	45%	0.80 (0.75–0.87)
Prenatal care	n = 531,442	n = 5646	
Zero visits	2%	5%	2.09 (1.77–2.46)
1–3 visits	8%	13%	1.56 (1.39–1.74)
4–6 visits	30%	40%	1.40 (1.30–1.51)
≥7 visits	60%	42%	reference
Gestation	n = 539,693	n = 5778	
Multiple	13%	12%	0.75 (0.68–0.84)
Delivery	n = 539,507	n = 5779	
Cesarean section	61%	63%	1.25 (1.17–1.35)
Sex	n = 539,824	n = 5782	
Male	53%	58%	1.27 (1.19–1.36)
Birthweight	n = 539,824	n = 5782	
<2500 g	49%	81%	3.43 (3.14–3.75)
1st- and 5th-minute Apgar	n = 423,470	n = 4725	
0–6 and 0–6	2%	24%	20.16 (18.44–22.04)
0–6 and 7–10	11%	26%	3.07 (2.83–3.33)
7–10 and 7–10	87%	50%	reference

## Data Availability

The data are publicly available just in part. The datasets were provided by Fundação SEADE. Requests to access these datasets should be directed to http://produtos.seade.gov.br/produtos/mrc/ (accessed on 7 March 2023). The database with linked information of birth and death certificates of São Paulo State, from 2004 to 2015, is available on request with the corresponding author.

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
