# Peer review of "Temporal Trend, Causes, and Timing of Neonatal Mortality of Moderate and Late Preterm Infants in São Paulo State, Brazil: A Population-Based Study"

_children, 2023, doi:10.3390/children10030536_

Round 1
Reviewer 1 Report
This manuscript addresses the important public health / epidemiologic problem of neonatal mortality of moderate and late preterm infants in a middle-income country - Brazil.
The stated objective of the paper was to analyze the temporal trend, causes and timing of neonatal mortality of infants born between 32-36 weeks gestation without congenital anomalies in the state of São Paulo, Brazil, from 2004 to 2015. This objective was adequately met and the authors addressed the more prevalent causes of mortality. However, a more robust discussion of how the causes can addressed and prevented should be provided in the discussion.
This is a population-based study as stated by the authors, but it was not immediately stated in the methods if the data was collected prospectively or retrospectively. This should be stated clearly in the materials and methods section.
Authors should provide a brief summary of method used for acquiring the data sets analyzed after the reference [22] provided. Were the data sets acquired yearly or en masse after 2015?
Some limitations of the study were described, are there any strengths?
Minor edit:
the words 'what are' in line 85 can be deleted.
Reviewer 2 Report
The article is well-written and grammatically correct and the content is well-organized.
The abstract gives a clear overview of the study including the research problem to be the object of the study, the way the sample was obtained and the findings.
I do not have any methodological concerns regarding this paper.
The majority of the literature used is of recent origin.
The findings are presented appropriately.
The discussion is well organized and addresses the research question and hypothesis posed in the introduction.
Author Response
We thank the reviewer for the careful reading of our manuscript.

Reviewer 3 Report
thank you
Reviewer 4 Report
Dear authors,
Thank you for the paper submitted covering an interesting topic. I think the article tries to analyze the temporal trend, causes and timing of neonatal mortality in infants who were born within 32-36 weeks gestation without congenital abnormalities in a state of Brazil during 2004 to 2015 I, however, have major comments and edits that I would like to suggest to the authors:
- Plagiarism: After using an anti-plagiarism tool, I have found that there are literally copied paragraphs and graphics from an article from the same investigation group whose similarity is 13%:
Guinsburg R, Sanudo A, Kiffer CRV, et al. Annual trend of neonatal mortality and its underlying causes: population-based study - São Paulo State, Brazil, 2004-2013. BMC Pediatr. 2021;21(1):54. Published 2021 Jan 26. doi:10.1186/s12887-021-02511-8
Furthermore, the conclusions in the abstract of this article are very similar to those presented in this peer review:
Conclusions from the article published in 2021: “Despite the significant decrease in neonatal mortality rate over the 10-year period in São Paulo State,
improved access to qualified health care is needed in order to avoid preventable neonatal deaths and increase
survival of infants that need more complex levels of assistance.”
Conclusions from the article to be published: “Although the neonatal mortality rate of infants 32-36 weeks gestation without congenital anomalies decreased significantly over the study period, improvement in the quality of perinatal care may further decrease these preventable deaths.”
Therefore, due to the lack of originality and novelty, I consider that this article does not provide new information in the field.
Line 236-238: I think the reference to the systematic review should be introduced.
I think a conclusion section should be added to the manuscript.

Reviewer 5 Report
The article concerns neonatal deaths in premature infants born in 2004-2015. The authors assessed e.g. annual trends of three main causes of them, i.e. asphyxia, respiratory disorders and infections. Were perinatal mortality analyzed in children born at the gestational age of 32.0-36.0 weeks (i.e. 36 weeks and 0 days) or 32.0-36.6 weeks (i.e. 36 weeks and 6 days)? It should be specified, because "term delivery" is defined from 37 weeks and 0 days.
Since the results concern only ONE state - Sao Paulo State - the outcomes cannot be generalized to the whole country. It is one of many regions where the level of medical services, access to hospitals and the level of education of patients certainly differs from other regions.
Please provide a brief description of this region, change the title, abstract.
The article lacks conclusions.
Since the Authors title the work as "addressing the challenge of preventanle neonatal deaths...", please summarize it in the conclusions.
Please list other causes of child deaths. Asphyxia, respiratory disorders and infections accounted for 85% of the causes in the study group. What were the rarer causes?
Reviewer 6 Report
To authors,
1. Line 91 around, you state the purpose (problem = clinical question) in this study, which should be used for the present paper title. The word “addressing” is unclear. For example, “Temporal tend, causes, and timing of neonatal mortality of infants born at 32-36 weeks in Brazil”.
2. The mortality rate was still high compared with UK or USA, I understand. Then, how about the “mortality-reducing rate”? It looks like Brazil showed “bigger/larger” reduction rate compared with UK or USA. Why? Is this simply because the original/initial high mortality rate gives “room” for large reduction thereafter?
3. Usually, death occurred due to “overlapping” causes. The present data was based on registration and thus there are some limitations to definitely determine the cause of death. Please touch this issue.
4. All three are of “categories” of preventable disease, I understand. Then, how do you intend to “utilize” the present data to reduce the mortality rate in the near future? You did not study it here. Thus, please very shortly 1) touch this issue OR 2) just state the following meaning “we did not study how many deaths were actually preventable in an individual-by-individual manner and thus we have no definite data for further reducing the mortality; however, the present study might provide a basic data for the future study to reduce neonatal mortality in this fraction”.
Round 2
Reviewer 4 Report
Dear authors,
Thank you for taking the time to address comments on the manuscript entitled:
“Temporal trend, causes, and timing of neonatal mortality of moderate and late preterm infants in São Paulo State, Brazil: a population-based study “.
The manuscript has been much improved, but there are still some issues about originality and novelty due to the publication of the precedent article entitled:
Guinsburg R, Sanudo A, Kiffer CRV, et al. Annual trend of neonatal mortality and its underlying causes: population-based study - São Paulo State, Brazil, 2004-2013. BMC Pediatr. 2021;21(1):54. Published 2021 Jan 26. doi:10.1186/s12887-021-02511-8
As you have already said, these two articles belong to the same family, but in my opinion, the precedent is 'the progenitor' and this is 'the son' that does not have any more information to present.
Therefore, due to the lack of originality and novelty, I consider that this article does not provide new information in the field and should be rejected.
Reviewer 5 Report
The Authors have adequately (and fully) addressed my concerns and suggestions in the revised version. Thank you :)